# Viruses in Wastewater—A Concern for Public Health and the Environment

**DOI:** 10.3390/microorganisms12071430

**Published:** 2024-07-14

**Authors:** Coralia Bleotu, Lilia Matei, Laura Denisa Dragu, Laura Georgiana Necula, Ioana Madalina Pitica, Mihaela Chivu-Economescu, Carmen Cristina Diaconu

**Affiliations:** 1Stefan S. Nicolau Institute of Virology, Romanian Academy, 030304 Bucharest, Romania; coralia.bleotu@virology.ro (C.B.); lilia.matei@virology.ro (L.M.); denisa.dragu@virology.ro (L.D.D.); laura.necula@virology.ro (L.G.N.); ioana.pitica@virology.ro (I.M.P.); carmen.diaconu@virology.ro (C.C.D.); 2Research Institute of the University of Bucharest (ICUB), University of Bucharest, 060023 Bucharest, Romania; 3The Academy of Romanian Scientist, 050711 Bucharest, Romania

**Keywords:** viruses, wastewater, wastewater treatment, environmental surveillance

## Abstract

Wastewater monitoring provides essential information about water quality and the degree of contamination. Monitoring these waters helps identify and manage risks to public health, prevent the spread of disease, and protect the environment. Standardizing the appropriate and most accurate methods for the isolation and identification of viruses in wastewater is necessary. This review aims to present the major classes of viruses in wastewater, as well as the methods of concentration, isolation, and identification of viruses in wastewater to assess public health risks and implement corrective measures to prevent and control viral infections. Last but not least, we propose to evaluate the current strategies in wastewater treatment as well as new alternative methods of water disinfection.

## 1. Introduction

Wastewater can contain various chemicals, microbiological pathogens, and other contaminants that can endanger human health and aquatic ecosystems [1,2]. Viruses have been cited as potentially the most important and hazardous pathogens found in wastewater. Wastewater can be contaminated from various human and non-human sources, which can vary depending on the local environment, wastewater management practices, and the level of development of a region [3,4,5]. Virus-contaminated wastewater can be discharged into the sewer system from toilets, sinks, and other sanitary installations, or industrial activities such as food production, chemical processing [6]. Recreational and tourist activities, wild and domestic animals, or agriculture represent other sources of contamination. Moreover, weather and natural conditions such as heavy rains or snowmelt and stormwater can carry contaminants from the ground surface and streets into sewers, contaminating water with viruses and other pathogens.

The wastewater viruses share certain characteristics and can cause infections ranging from asymptomatic, mild forms to severe gastroenteritis, conjunctivitis, hepatitis, respiratory disease, and central nervous system infections [7]. Some viruses, such as noroviruses and rotaviruses, are highly contagious and can cause rapid disease outbreaks in communities. 

Of great importance is the fact that viruses can survive in aquatic environments for long periods of time and can be resistant to conventional water disinfection treatments. Several factors, including water composition, temperature, and exposure to light, influence the survival of viruses in water. Environmental factors such as ultraviolet light and temperature play a significant role in the survival of viruses in water, with exposure to sunlight accelerating the inactivation of viruses, while lower temperatures can prolong their survival [8]. Enteric viruses, such as polio-, echo-, and coxsackieviruses, can survive in freshwater sources for long periods, with average inactivation rates ranging from 0.174 log10 d^−1^ in groundwater to 0.576 log10 d^−1^ in surface water at the tap [8]. Rotaviruses, known to cause diarrhea in children, can persist in freshwater sources for approximately 10 days at 20 °C, and 32 days at 4 °C [9]. On the other hand, in municipally treated tap water, the rotavirus titer did not decrease at 4 °C for 64 days, while, at 20 °C, it decreased by about 2 log10 during the same period, with data indicating that contamination with rotavirus intervened after water chlorination [9]. Enteric adenoviruses were more stable than other enteric viruses in tap water at various temperatures for up to 60 days. For example, at 4 °C, poliovirus and hepatitis A virus decreased by 1.6 and 2.7 log10, respectively, while adenovirus type 40 decreased by less than 0.5 log10 and type 41 by almost 1 log10, which revealed differences in virus survival by type and temperature [10]. Other research compared the survival of hepatitis A virus and poliovirus in bottled mineral water at 4 °C and room temperature, finding that each virus persisted longer at 4 °C with little inactivation for over 1 year, but, nevertheless, at room temperature, differences in virus survival were very evident, with the longer survival of hepatitis A virus [11]. In conclusion, some viruses present in wastewater can survive environmental conditions and wastewater treatment processes, persisting in the aquatic environment and infecting humans and other organisms. This makes wastewater viruses a significant threat to human health. To effectively manage the risks associated with viruses in wastewater, it is essential to adopt appropriate wastewater treatment measures, as well as implement appropriate hygiene and public health practices to minimize human exposure and environmental impact.

This review aims to present the major classes of viruses in wastewater, as well as the methods of concentration, isolation, and identification of these wastewater contaminants to assess public health risks and implement corrective measures to prevent and control viral infections. Last but not least, we propose to evaluate the current strategies in wastewater treatment as well as new alternative methods of water disinfection.

## 2. The Major Viruses in Wastewater

Contaminated wastewater serves as a significant reservoir for viruses, representing one of the major ways of microbial transmission, along with the fecal–oral route and person-to-person contact. The presence of viruses in wastewater can vary by region (geographical and climatic characteristics), development level, population health status, and wastewater management practices. Wastewater surveillance of pathogenic viruses is used to monitor the circulation of the different viruses in a population and to predict disease outbreaks (Figure 1).

Enteric viruses, adenovirus, rotavirus, norovirus, enterovirus, and hepatitis A virus are the most prevalent viruses in wastewater due to their high excretion from infected individuals, low removal in wastewater treatment processes, long-term survival in the environment, and low infectious dose. 

*Human adenoviruses* (HAdV) include 88 different genotypes [12] that cause a variety of diseases, such as pneumonia, respiratory tract infections, conjunctivitis, and gastroenteritis [13]. The infections in the gastrointestinal tract are frequently associated with genotypes 40 and 41. Epidemiological studies have shown that HAdVs are highly frequent in wastewater in Taiwan (27.3%), Morocco (45.5%), Tunisia (64%), Poland (92.1%), and Norway (92%) [14,15,16,17,18]. In April 2022, cases of fulminant hepatitis induced by Adenovirus 41, in immunosuppressed patients, were reported in Scotland. Later, cases were reported in several parts of the world, leading to the “2022 pediatric hepatitis outbreak” [19,20,21]. One of the hypotheses suggested then was that, considering the survival of the adenovirus in sewage water and its resistance to the treatments applied to obtain recycled sewage water that developed countries use for consumption and for the irrigation of crops, it is possible that the link between these new hepatitis cases is recycled sewage water [22].

Because of their high frequency, prevalence, and persistence in aquatic environments, HAdVs can serve as index organisms for viral pathogens to monitor viral pollution in the context of wastewater monitoring [4,23].

Another important viral agent identified in wastewater is Rotavirus (RV), which is the most important cause of severe diarrhea among young children and infants [24]. According to the WHO, in 2004, RVs caused over 500,000 deaths worldwide, mainly in developing countries. Its prevalence in wastewater was evaluated in several studies, being detected in 10% of the samples in Uruguay [25], 25% in Iran [26], 50% in Thailand [27], and 90% in Spain [28]. Rotaviruses are small viruses, very resistant to wastewater treatment, that can survive in the environment for extended periods; therefore, the WHO recommended, in the Guidelines for drinking-water quality, the use of RV as a reference pathogen [29].

Noroviruses (NoVs) are known as principal agents of acute gastroenteritis worldwide, being responsible for 685 million infections annually according to the WHO [30] and are associated with almost 50% of all-cause acute gastroenteritis outbreaks [31,32]. The majority of causes of NoVs gastroenteritis are associated with the GII.4 genotype, associated with a more severe disease in people of all age groups [33,34,35]. Noroviruses are commonly studied and detected in wastewater and have also been extensively investigated for their presence in human feces. Their high contagiousness is attributed to a low infectious dose (requiring 18–2800 viral particles to infect a healthy adult), and, together with the prolonged duration of viral shedding, NoVs can lead to large outbreaks. According to a recent meta-analysis, the overall detection rate of NoVs in wastewater was 82.10%, with a higher concentration in spring and winter [36].

Human enteroviruses (EVs) comprise several medically important viruses that are currently monitored in wastewater. These include the hepatitis E virus (HEV), polioviruses, enterocytopathic human orphan (ECHO) virus, coxsackieviruses, and human enteroviruses (68–121) with more than 100 serotypes [37]. They may cause a wide range of diseases, including gastroenteritis, conjunctivitis, herpangina, myocarditis, hepatitis, poliomyelitis, and meningitis. Regarding hepatitis, it was shown that hepatitis A virus (HAV), along with hepatitis E virus (HEV)**,** is linked to wastewater contamination and poses significant public health risks due to their ability to survive and spread through the fecal–oral route. The virus is shed in the stools of infected persons, and it is transmitted mostly by contaminated drinking water, causing large outbreaks as well as sporadic cases. In recent years, the global monitoring of HAV frequency in wastewater has shown lower occurrences in high-income countries in Europe, ranging from 3.1% to 60%. Conversely, countries with lower living standards or higher endemicity, such as South Africa, Cairo, and Tunisia, have reported higher detection frequencies (>60%) [38]. Moreover, HEV, the etiological agent of hepatitis E, is responsible for over 20 million cases of infections, with more than 40,000 deaths annually according to the WHO [39]. Wastewater testing can provide valuable information on HEV circulation in endemic areas and serve as an early warning tool for disease outbreaks that could lead to a faster and improved public health response.

Poliovirus and non-polio EVs wastewater assessment plays an important role in exploring the circulation of these viruses in the community. In fact, at least 30% of concentrated sewage from grab samples and at least 10% of traps should reveal non-polio enterovirus [40].

The first attempts to isolate the poliovirus from the sewage were made in 1939 [41], and, since then, scientists have conducted studies to monitor the presence of various human pathogens in the sewage of the cities [42,43,44,45,46]. Moreover, environmental surveillance has been used as a supplementary tool to monitor the circulation of poliovirus in specific populations since 2013, when the World Health Organization (WHO) included the environmental surveillance of poliovirus in the new Strategic Plan of the Global Polio Eradication [40,45]. Recently, in several polio-free countries, wastewater testing allowed the detection of vaccine-derived poliovirus and revealed its community transmission [47,48,49,50], emphasizing once again the importance of environmental surveillance.

Similarly, during the COVID-19 pandemic, environmental surveillance has been used to detect the presence of severe acute respiratory syndrome coronavirus 2 (SARS-CoV-2) in wastewater, and it proved to be a valuable indicator of infection in a community. More than that, it has been shown that the viral concentration in wastewater correlates with the number of COVID-19 cases [51], and it seems that the appearance of the virus in the wastewater precedes the clinical data by 4–10 days [52]. Many countries have developed projects for monitoring SARS-CoV-2 RNA in wastewater in order to predict outbreaks of COVID-19. Worldwide, there are more than 70 countries that are implementing wastewater surveillance for SARS-CoV-2 and, lately, also for other respiratory viruses such as influenza and respiratory syncytial virus (RSV). Thus, public health programs carried out in the last decades by most countries have proven that wastewater monitoring can represent a powerful early warning system against viral pathogens. Enteric viruses are among the commonest human pathogens found in wastewater, being responsible for the occurrence of sporadic cases as well as outbreaks [53].

In addition to commonly identified viruses, studies have revealed, less frequently, the presence of other viruses in sewage, as their mode of transmission does not involve the fecal–oral route. One of the viruses less commonly found in wastewater is Ebola virus which causes severe hemorrhagic fevers in humans and animals. Although transmission occurs through direct contact with infected bodily fluids, such as blood and secretions, it has been shown that the virus persisted longer than expected in the wastewater environment [54]; therefore, the possibility of indirect infections after exposure to contaminated wastewater cannot be excluded [55]. Other non-waterborne viruses that have been detected in both human feces and wastewater, along with other zoonotic viruses, are Hantavirus, Dengue virus, Yellow fever virus, and Nipah virus [56]. Moreover, vector-borne viruses transmitted by insects such as West Nile virus or Zika virus that breed in water have been found in wastewater, together with Rift Valley fever virus and Chikungunya virus [57]. Another study confirmed the presence of Zika virus in wastewater [58].

Human immunodeficiency virus (HIV) was found in 1992 by Casson and his colleagues in wastewater. They demonstrated that the virus survives for 12 h in wastewater, but loses its infectivity in the following 48 h [59]. However, current studies have not confirmed the presence of the HIV virus in wastewater samples [60].

These viruses can be transmitted by contacting contaminated water or ingesting contaminated water or food (Table 1).

Although epidemiology provides an essential basis for understanding health outcomes and the burden of disease by measuring the distribution and determinants of health events, being of particular value in estimating the incidence of certain diseases in the population and determining the proportion of cases attributable to different exposures, these studies are not always applicable to individual local sources due to the small population size and insufficient number of cases to ensure sufficient statistical power. Therefore, the Quantitative Microbial Risk Assessment (QMRA) and other risk assessment techniques are more feasible and often the only approach available to water station managers to assess the risk to consumers. The QMRA assesses potential risks to human health and is complementary to epidemiological studies for water safety management [72]. The QMRA consists of four successive steps: (a) the identification of potentially dangerous pathogens; (b) estimating the dose of pathogens to which individuals are exposed in a certain scenario; (c) the analysis of the relationship between the dose of pathogens and the probability of infection or disease; and (d) risk characterization by assessing the probability of infection and the impact on health. To calculate the risk associated with the exposure to environmental microbial pathogens, the World Health Organization (WHO) recommends using the QMRA combined with a Monte Carlo simulation, which helps assess the variability and uncertainty in risk estimates. To measure the burden of disease, the QMRA uses the disability-adjusted life years (DALY) metric, which includes years of life lost (due to premature mortality) and years lived with disability (due to diseases and health conditions) [73]. The various choices of the dose–response model highly influenced the health risks [74]. The QMRA analyzes and allows authorities to make informed management decisions to implement specific measures to reduce occupational health risks for wastewater treatment plant (WWTP) workers.

## 3. Methods and Laboratory Techniques for Spotlighting Viruses in Wastewater

Virus isolation from wastewater is an essential process in controlling water quality and preventing the spread of viral infections. Methods for isolating viruses from wastewater may vary depending on the available technology and the specific objectives of the analysis. The common steps of virus isolation methods from wastewater include the following: (1) collecting representative samples from the wastewater source in sterile containers for collection and following safety protocols, as viruses can be contagious and pose a risk to laboratory personnel; (2) the concentration of viruses, because wastewater contains a low concentration of viruses; (3) the extraction of viruses by different methods, such as precipitation with organic solvents or using mechanical resuspension; (4) the analysis of viruses using specific methods to identify and quantify specific types of viruses (e.g., polymerase chain reaction (PCR) or culture cells to demonstrate the infectivity of the virus); and (5) interpreting the results, which can provide information about the presence, concentration, and types of viruses in the wastewater. This information can be used to assess health risks and take appropriate measures to treat or eliminate viruses. It is important to note that the exact methods for isolating viruses from wastewater may vary depending on the laboratory and the specific objectives of the analysis.

### 3.1. Methods for Wastewater Viruses’ Concentration

The effectiveness of virus detection relies on factors such as sample volume, yield of nucleic acid extraction, and purity. Raw wastewater samples typically contain higher levels of viruses than other environmental samples due to their higher turbidity, suspended solids, and organic matter concentrations. Additionally, influent wastewaters often contain high levels of humic acids and heavy metals, which can interfere with virus detection methods. Sludges, being heterogeneous matrices, pose challenges as viruses tend to adsorb onto the surface of flocs. Consequently, concentration steps are necessary, including primary and secondary concentration methods. It is important to note that molecular detection methods for viruses do not indicate their infectivity but only detect molecular fragments (DNA or RNA). Assessing infectivity requires maintaining virus viability throughout sampling and processing. 

#### 3.1.1. The Virus Adsorption–Elution (VIRADEL) 

The virus adsorption–elution (VIRADEL) method involves removing viruses from water samples by adsorbing them onto a membrane filter, and then eluting them for recovery. Initially developed by Metcalf in 1961 [75] and further refined by Wallis and Melnick in 1967 [76], this method relies on the electrostatic attraction between viral particles and the membrane surface. While originally designed for low-turbidity water samples [77], adaptations have been made for high-turbidity samples like influent wastewater, which may require pre-filtration. VIRADEL has been extensively used for processing large volumes of samples ranging from 0.5 to 400 L. from various sources such as rivers, tap water, groundwater, and coastal waters [78]. However, the increase in the volume of samples is correlated with the increase in the concentration of PCR inhibitors, necessitating a careful consideration of the sample volume to ensure a high viral content and low inhibitor levels. Despite its effectiveness, VIRADEL has drawbacks, including a multi-step process (pre-filtration (electronegative or electropositive filtration), membrane adsorption, and elution), resulting in a prolonged processing time, and reduced virus recovery yields [78,79]. Electronegative filtration, a common approach in VIRADEL methods, involves adjusting the pH of the sample to induce a charge inversion on the surface of viruses (negatively charged), allowing them to be adsorbed onto negatively charged membranes [76,80]. This method, first explored by Wallis and Melnick in 1967, has been refined over the years, with the addition of salts like magnesium or aluminum chlorides to enhance viral adsorption [78]. The recovery of viruses is typically achieved through elution from the membrane for further analysis. For negatively charged filter membranes, elution solutions often include an alkaline solution of beef extract in glycine buffer, which restores the virus’s negative charge, causing it to repel from the filter [80]. Other elution buffers like glycine/NaOH, sodium polyphosphate, or skimmed milk have also been used to recover viruses from environmental samples [78]. Following elution, the eluate often undergoes secondary concentration to reduce the sample volume before nucleic acid extraction. An alternative approach, electropositive filtration, utilizes membranes with a positive charge, eliminating the need to adjust the sample pH for virus adsorption [81]. Studies have compared the efficiency of different electropositive filters in concentrating viruses from wastewater treatment plant effluents, with varying success rates. Some electropositive filters were developed to increase the effectiveness of wastewater virus concentration: nano-alumina fibers (NanoCeram filter, Argonide Corporation, Sanford, FL, USA) [82,83], filter cartridge system with electropositive AlCl_3_ granule [84], glass and cellulose medium (Virosorb 1MDS filter, CUNO, part of 3M Separation and Purification Sciences Division, Meriden, CT, USA) [83,85], nanodiamond-coated quartz microfiber membranes [86], etc. 

#### 3.1.2. Ultrafiltration

Ultrafiltration methods employ membrane filters with pore sizes smaller than viral particles, allowing for size-based exclusion. A disadvantage of ultrafiltration is the potential for the clogging of the small pore filter when dealing with turbid samples, but tangential flow ultrafiltration can solve this problem by allowing water to flow parallel to the membrane surface, making it suitable for raised-turbidity samples [77,78].

Through the centrifugal ultrafiltration method, the sample volume used is greatly reduced with a recovery rate comparable to VIRADEL. Thus, from the 10 L used at VIRADEL, the volume was reduced to 100 mL in the method developed by Qiu et al. (2016), which uses a Centricon Plus-70 centrifugal filter with a molecular weight of 30 kDa to concentrate human enteric viruses from wastewater in a single step [79], or to 10 mL of wastewater samples in the method of Sidhu et al. (2013) who used centrifugal ultrafiltration to concentrate human adenovirus [87]. Unlike electronegative filtration methods, centrifugal ultrafiltration does not necessitate pH adjustments before or after filtration, preserving the stability of pH-sensitive viruses, and offers advantages such as easier sample collection, simpler processing, shorter filtration times, and the ability to process multiple samples simultaneously [78].

#### 3.1.3. Direct Precipitation with Polyethene Glycol (PEG)

Direct precipitation with polyethene glycol is a common method used to concentrate viruses from environmental samples. PEG, an inert and biocompatible polymer, acts by sequestering water molecules from the solvation layer surrounding viral capsids, enhancing virus–virus interactions and resulting in precipitation [78,88]. Modified PEG methods, like Pyro-PEG, have been developed to increase the virus detection efficiency by incorporating additional steps like elution with sodium pyrophosphate and sonication [89]. While PEG is efficient, it may cause non-selective precipitation, including various interfering proteins, potentially hindering subsequent PCR-based detection methods [90,91]. The removal of inhibitors by a guanidinium isothiocyanate (GIT) extraction to purify and precipitate viral nucleic acid after PEG extraction increases the sensitivity of virus detection both in cell cultures and by PCR in samples initially considered negative [91]. 

#### 3.1.4. The Skimmed Milk Flocculation Technique

The skimmed milk flocculation technique involves three key steps: (i) the adsorption of viruses to pre-flocculated skimmed milk proteins, (ii) the sedimentation of virus-laden flocs, and (iii) the dissolution of sediments using a phosphate buffer solution [78]. The method was successfully used for the recovery of adenovirus from sludge [92], adenovirus (HAdV 35), rotavirus (RoV SA-11) [93], SARS-CoV-2 [94], etc. This technique offers advantages such as minimal equipment requirements and a reduced number of processing steps, facilitating the simultaneous concentration of numerous samples [95].

Another method, the *pre-treatment of sludge samples*, is essential for effectively recovering viruses due to their adsorption onto sludge flocs through electrostatic and hydrophobic interactions. The traditional method involves desorbing viruses from sludge flocs using an AlCl_3_ solution (pH 3.5), followed by elution and filtration. Sonication can improve viral desorption, but excessive sonication could damage viral particles [96]. Moreover, using non-ionic surfactants, such as Tween-80, to dislodge viruses from the sludge matrix can enhance virus counting efficiency [97]. Enzymatic virus elution (EVE), using 10 g/L hydrolytic enzymes (lysozyme, carboxylesterase, chymotrypsin, and papain) and a cation exchange resin, also reduces PCR inhibitors and increases virus recovery efficiency [98]. A comparison between beef extract and glycine elution indicated that lower organic matter levels with glycine led to improved PCR amplification efficiency [99]. These advancements highlight the ongoing efforts to optimize pre-treatment methods for efficient virus recovery from sludge samples. Despite efforts, methods like ultracentrifugation showed limited virus recovery and PCR inhibition reduction. 

#### 3.1.5. Ultracentrifugation

Ultracentrifugation is a sophisticated and efficient method for concentrating viruses from various media, including wastewater. This method separates and concentrates viral particles from a liquid suspension by applying very high centrifugal forces (typically over 100,000× *g*) to the samples, causing the components to separate according to their density. Dense particles, such as viruses, are forced to the bottom of the tube, while less dense components remain suspended in the supernatant. Ultracentrifugation showed limited virus recovery and PCR inhibition reduction. A study comparing two viral concentration methods, ultracentrifugation and skimmed-milk flocculation, for the detection of SARS-CoV-2 in wastewater samples demonstrated that the ultracentrifugation method displayed a higher analytical sensitivity for the detection of enveloped viruses [100]. Another study revealed that ultracentrifugation can be considered a suitable method to concentrate viruses directly from wastewater with a recovery percentage between 66 and 72% [101]. However, in the case of this method, the efficiency of the results depends on the specific density, morphology, and membrane attachment characteristics of each virus [102]. Although it involves cost and operational complexity, its advantages in terms of purity and efficiency make it a preferred choice in virus research and analysis laboratories.

The major advantages and disadvantages that must be considered to determine which method is appropriate for a given context or specific application of wastewater virus concentration are presented in the Table 2.

### 3.2. Methods for Wastewater Viruses’ Identification

The samples are processed differently depending on the method of detecting viruses in wastewater samples. 

#### 3.2.1. PCR

In PCR-based methods, the target sequence of the virus’s genetic material (RNA or DNA) is amplified, and the quality and purity of these biomacromolecules affect the efficiency of the amplification and quantification methods. In the case of RNA viruses, a reverse-transcriptase step is needed before PCR amplification. This method is extremely useful in the case of viruses that cannot be cultivated in cell cultures or that are relatively noncytopathogenic for cultured cells. Moreover, using the PCR technique, the time needed for viral detection is reduced. For example, in the case of enterovirus detection in water, the time needed for analysis varies between 14 days using cell culture and less than a day using PCR [103]. Real-time PCR, a variant of conventional PCR, allows the evaluation of viral genome concentrations and represents an important technique for the detection of environmental viruses. During each cycle, the amplicon can be quantified using SYBR Green (nonspecific attachment to double-stranded DNA), or a fluorescent internal probe (specific hybridization) [104]. Conventional PCR/RT-PCR and quantitative real-time PCR (qPCR) were used for the detection of rotavirus A (RV-A), human adenoviruses (HAdV), norovirus genogroup I and II (NoV GI/GII), and hepatitis A viruses (HAV) in sewage samples from hospital wastewater treatment plants, confirming the potential for environmental contamination by viruses, and suggested the necessity of establishing new standards for policies on wastewater management [105].

#### 3.2.2. ddPCR

With the advancement in technology, new tools that allow the quantification and precise identification of viruses, even in samples with low concentrations of viruses, such as digital (dPCR) platforms, have started to be used more and more often. Two primary partitioning techniques are utilized: droplet-based and chip-based/microfluidic. Droplet digital PCR (ddPCR) employs water–oil emulsion to generate picoliter droplets, with platforms like QX100™/200™ ddPCR Systems (Bio-Rad Laboratories, Hercules, CA, USA) and Naica^®^ System (Stilla Technologies, Villejuif, France). This technique involves manual or automated droplet generation, followed by the thermocycling and reading of partitions using dedicated instrumentation. Each reaction typically produces 10,000 to 20,000 droplets [106]. Chip-based/microfluidic partitioning, on the other hand, uses nano-fabricated nanoliter reaction chambers. Platforms like QuantStudio™ Absolute Q™ (Applied Biosystems, Waltham, MA, USA), QIAcuity (QIAGEN, Hilden, Germany), and BioMark™ HD (Fluidigm, South San Francisco, CA, USA) employ this technique. Reactions are partitioned into fixed micro-chambers, allowing for the precise quantification of genetic targets. Each platform has its unique workflow, from reaction preparation to partitioning and data analysis [106]. The digital droplet PCR method enabled the accurate detection of noroviruses in a complex environment such as wastewater, which is essential for monitoring the risk of virus contamination in water sources [107,108,109].

To monitor the presence of viruses in wastewater in order to obtain important information about the spread of diseases in communities and to guide public health measures, droplet PCR was used during respiratory pandemics produced by viruses such as influenza virus [110,111], respiratory syncytial virus (RSV) [112], and SARS-CoV-2 [113,114,115,116,117]. Enteric viruses, such as rotaviruses and adenoviruses, present in wastewater have also been quantified using dPCR, such as Rotavirus and Enterovirus in untreated sewage [118,119].

Therefore, dPCR is frequently used to detect viruses in wastewater in water-quality-monitoring studies and epidemiological research, as it is a rapid and reliable method to assess the presence of pathogenic viruses in aquatic environments, identifying potential foci of infection and evaluating the effectiveness of water treatment and sanitation measures. dPCR platforms offer versatile and efficient tools for accurately quantifying genetic targets in water, with each platform having strengths and limitations. However, dPCR results are more influenced by extraction methods, reaction chemistries, and sample concentration than by platform variation [106]. On the other hand, the use of droplet PCR can be limited by cost and the need for specialized equipment and technical expertise. However, with the advancement in technology and the development of more affordable kits and protocols, it is expected that droplet PCR will become an increasingly widespread and accessible method for the detection of viruses in wastewater. Currently, due to the vast diversity of micro-organisms and their varied characteristics, and also the limitations related to isolation and concentration methods, there is no single method that allows the identification of all micro-organisms present in water samples. 

#### 3.2.3. Next-Generation Sequencing (NGS)

NGS has been used as a robust and efficient approach for detecting and monitoring pathogens in wastewater, enabling broad community surveillance and identifying public health threats. There are two main types of sequencing technologies: short-read and long-read. Short-read sequencing technologies (such as those from Illumina (San Diego, CA, USA), Ion Torrent (Guilford, CT, USA), and Pyrosequencing (Isleworth, UK)) produce millions of short reads in each run with the advantage of a high accuracy in determining the sequence of nucleotide bases and sequencing speed, allowing efficient and rapid analyses. The disadvantages of short-read sequencing are related to the length of relatively short fragments (between 50 and 300 bases), which can limit the ability to detect large genetic variations or complex structures, as well as the difficulty in assembling the genome because short fragments can lead to ambiguous overlaps [120]. In contrast, long-read sequencing (e.g., Pacific BioSciences (Menlo Park, CA, USA) and Oxford Nanopore (Oxford, UK)) generates much longer reads, up to 20 kilobases, without requiring assembly, which are valuable for understanding the genetic context and precisely detecting the origin of genes. However, they have higher error rates and higher initial costs [120]. There are four main subcategories of NGS with their specific methods and workflows: whole-genome sequencing (WGS—used mostly for characterizing micro-organisms isolated in culture, usually using short-read technologies and de novo assembly or guided assembly to construct the whole-genome sequence), metagenomic sequencing (involves the random sequencing and subsampling of genetic material from an environmental sample or mixed community), metatranscriptomic sequencing (used to identify RNA viruses and to obtain information about the genetic activity and functionality of micro-organisms), and targeted sequencing of amplified gene regions (amplicon sequencing) [120]. Compared to traditional technologies, NGS enables the rapid and efficient sequencing of a large number of DNA or RNA molecules in a single experiment, with the high sensitivity allowing the detection of small amounts of genetic material or subtle genetic changes.

A higher sensitivity of sequencing than RT-qPCR was observed in the detection of viruses such as Enterovirus spp., influenza A and B viruses, and RSV (Enteroviruses were detected in all samples tested using sequencing, while only one sample was positive using RT-qPCR; influenza A virus was detected in 75% of samples using both sequencing and RT-qPCR, but only 50% of samples overlapped in successful detection in both methods; influenza B virus was not detected by RT-qPCR, but was successfully detected in 63% of samples using sequencing. RSV was more difficult to detect by both methods: all samples were negative by RT-qPCR tests, but 25% were positive by sequencing) [121]. On the contrary, a study comparing targeted amplicon sequencing for SARS-CoV-2 (ARTIC v3 protocol) with RT-ddPCR quantification for detecting five mutations characteristic of variants of concern (VoCs) in 547 wastewater samples observed positive 42.6% mutation detections by RT-ddPCR that were missed by sequencing, due to negative detection or the limited read coverage at the mutation position. Further, when sequencing reported negative or depth-limited mutation detections, 26.7% of those events were instead positive by RT-ddPCR detections, highlighting the relatively poor sensitivity of sequencing. No or weak associations were observed between quantitative measurements of target mutations determined by RT-ddPCR and sequencing. These findings caution against using quantitative measurements of SARS-CoV-2 variants in wastewater samples determined solely based on sequencing [122].

On the other hand, a study that investigated the presence, prevalence, and diversity of human adenoviruses (HAdVs) in wastewater (22 treatment plants from 10 regions in Italy), observing a prevalence of HAdVs of 60.3%, showed that, through Sanger sequencing, only four species and four different types [A (type 12), B (type 3), C (type 5), and F (type 41)] were identified, and, by NGS, an additional 16 types were detected [types 12 and 31 (species A), type 3 (species B), types 1, 2, and 5 (species C), types 9, 17, 24, 26, 37, 38, 42, 44, 48, and 70 (species D), type 4 (species E), and types 40 and 41 (species F)] [123]. The results underline the diverse circulation of HAdVs in raw wastewater and confirm the efficiency of NGS in detecting less prevalent types.

The targeted amplicon sequencing approach is a gold standard for identifying and analyzing variants. However, when applied to environmental samples such as wastewater, it remains unclear how sensitive this method is for detecting variant-associated mutations in environmental samples. The disadvantages of NGS technology are related to the high initial costs of the required equipment and IT infrastructure for data analysis, with complex bioinformatics analysis and considerable computational resources required to interpret and extract relevant information. Technical errors and biases associated with the sequencing process can affect data quality and interpretation, requiring further validation and correction. 

There are several NGS amplicon detection panels for respiratory and enteric viruses. For example, the Viral Surveillance Panel (Illumina) allows the characterization of more than 200 viruses and subtypes that are of high risk to public health, including SARS-CoV-2, Influenza, Mpox Virus, and Poliovirus. These panels are compatible with a range of samples, including wastewater, and can detect both RNA and DNA viruses.

#### 3.2.4. DNA Microarrays, DNA Arrays, Gene Chips, or Lab-on-Chip

DNA microarrays, DNA arrays, gene chips, or lab-on-chip, a miniaturized technology where thousands or tens of thousands of DNA probes are immobilized on a special glass or silicon chip with a surface area of only a few square centimeters [124,125], offer a high-throughput and multiplexed approach for analyzing microbial communities in samples, allowing the simultaneous analysis of many samples for the presence of specific virus sequences or detecting polymicrobial pathogens from a single specimen. According to Dufva (2005), the array geometry, spot density, spot characteristics (morphology, probe density, and hybridized density), background, specificity, and sensitivity are indicators that measure the performance of the technique [126]. 

It is a technique with a high specificity, able to provide valuable insights into microbial diversity and dynamics. There are some studies [127,128] that report the detection of viruses present in wastewater samples using the microarray technique, most of the time coupled with another technique (e.g., RT-PCR). It is important to mention that there are numerous limitations encountered related to the validation of the signals, the non-specificity probes, the possibility of cross-hybridization with non-targeted viruses, scanning errors, and errors in the generation of clones in the stages before printing on slides [127,129,130,131,132]. Over time, each of these limitations has challenged researchers to carry out new studies to find solutions. Microarray techniques offer a low sensitivity compared to PCR methods [127,133], especially when the concentration of viruses is very low, but, on the other hand, they offer increased specificity, identifying viruses that PCR methods may fail be identified, as validated by sequencing [127]. As proposed by Wong et al. (2013), the microarray technique, coupled with PCR methods, remains a suitable method for screening community wastewater for a broad number of pathogenic viruses [127]. It is a technique that cannot be used in the discovery of a new virus, but, using microarray elements derived from highly conserved regions within viral families, it can be used in the discovery of the novel variants of known viruses [134]. Regarding the cost and the field of application, microbial detection arrays are in between multiplex PCR (low-cost, narrowly focused on a very specific target) and high-throughput sequencing [135]. 

Initially, in 2002, Wang et al. (2002) designed a first-generation viral detection microarray to detect multiple viruses in human respiratory specimens. This microarray contained the most highly conserved 70 mer sequences from every fully sequenced reference viral genome in GenBank. In 2003, during the outbreak of severe acute respiratory syndrome (SARS) in March 2003, this technique contributed to identifying the presence of a previously uncharacterized coronavirus in a viral isolate cultivated from a SARS patient and subsequently confirmed the ViroChip’s finding by sequencing this genome of the virus [136,137]. Since then, the Virochip has been ported to an Agilent microarray platform and consists of ~36,000 probes derived from over ~1500 viruses [138]. Also using the Agilent inkjet deposition system (Agilent Technologies, Santa Clara, CA, USA), the GreeneChipPm, a pan microbial microarray comprising 29,455 sixty-mer oligonucleotide probes for vertebrate viruses, bacteria, fungi, and parasites, was designed by Palacios et al. (2007) [133]. The system introduces sample preparation and labeling methods that enhance sensitivity and, also, the analytical programs for data processing are more user-friendly to facilitate clinical application. GreeneChipVr (only for viruses) contained 9477 probes to address all vertebrate viruses (1710 species, including all reported isolates) [133].

Currently, numerous systems have been reported for the detection of viral pathogens using microarray technology: SMAvirusChip surveillance of arboviruses [139]; Damin et al. (2021) have developed CovidArray [140], and the FluChip-55—an influenza microarray [141], but also pan-Microbial Detection Array, and Lawrence Livermore Microbial Detection Array to detect all known viruses (including phages), bacteria, and plasmids [142]. 

Integrating microarray analysis with other techniques, such as PCR, next-generation sequencing, and traditional culture-based methods, can provide a more comprehensive understanding of microbial communities in wastewater.

#### 3.2.5. Electron Microscopy

Although all viruses are too small for detection using light microscopy, the direct visualization of these particles in wastewater samples with detailed images of the structure and morphology of viruses can be obtained using electron microscopy technology. Moreover, transmission electron microscopy (TEM) represents one of the first methods used for the quantification, identification, and classification of viruses according to their morphology and the only imaging technique that allows the direct visualization of viruses. Virus ultrastructure can be highlighted even using the standard protocols of TEM that include negative staining and ultrathin sectioning [78,143]. TEM was considered a powerful diagnostic tool for the detection of viral infection in biological samples, either in suspension or in tissue sections [143]. Even if currently advanced genomics and proteomics techniques are being developed for the detection of viral infections, TEM has the advantage of being able to identify viruses with a high precision, even in the case of viruses that have undergone mutations at the genome level [144]. Alhamlan et al. (2013) used TEM to highlight the presence of viral-like particles in lagoon waters prior to subsequent metagenomic analyses, noting variable morphologies such as tailed and filamentous forms [145]. One of the disadvantages is the impossibility of accurately quantifying the viral load and the difficulty of analyzing a large number of samples due to the fact that this technique cannot be automated [97]. 

#### 3.2.6. Staining of Viral Nucleic Acids Using Fluorochromes

For the quantification of viruses, the staining of nucleic acids using fluorescent dyes after the water samples are passed through a filter with a pore size of 0.22 μm followed by visualization with an epifluorescence microscope has been successfully used [96,146], having the advantage that stained virus particles can be counted even at lower magnifications, thus eliminating the need to use TEM instrumentation [147], and, on the other hand, it seems to allow the quantification of viruses that cannot be cultured in the laboratory but without distinguishing between them [78]. The combination of the use of specific fluorescent dyes for nucleic acids and flow cytometry was also applied for the quantification of viruses in wastewater samples, the method in which colored particles pass in a continuous flow through a laser slit and the emitted fluorescence is quantified in correlation with the DNA/RNA content of the sample [97,148,149]. In comparison with epifluorescence microscopy, flowcytometry appears to quantify a greater number of viruses, being more sensitive and also much faster [148,150]. The challenges and limitations of using flow cytometry to detect viruses in natural and engineered environments are related to the correct stability of the acquisition settings and depending on the appropriate controls to demonstrate that these signals are indeed from the virus and the sensitivity and accuracy of the instruments [151]. The advantages are related to the high speed of quantification, but, although it seems efficient to count viruses using these methods, they cannot identify viruses as such.

#### 3.2.7. Detection of Viral Antigen

The immunofluorescence (IFA) and enzyme-linked immunosorbent assay (ELISA) are two important techniques that have been used for the detection and quantification of viruses in wastewater samples that rely on the binding of the antigen to its specific antibody. The antigen–antibody complex appears as a fluorescent particle when the secondary antibody is fluorescently labelled and is observed under a fluorescent microscope (immunofluorescence) or a color reaction is produced (ELISA). Immunofluorescence was successfully used to quantify human adenovirus (HAdV) and JC polyomavirus (JCPyV) in untreated wastewater and showed an order-of-magnitude-higher sensitivity than other cell culture methods [152,153]. ELISA was used to detect rotavirus A (RoV-A) in wastewater samples, allowing the determination of virus removal efficiency from urban and hospital wastewater treatment systems [26]. The two techniques offer sensitive methods for monitoring viruses in wastewater but are quite expensive, requiring primary antibodies specific to a virus of interest and secondary antibodies labelled fluorescently or with an enzyme substrate. The methods cannot be used for the global identification of existing viruses in wastewater due to the fact that it is necessary to use specific viral antibodies, but they can be essential for the specific monitoring and evaluation of the effectiveness of wastewater treatment systems in eliminating viruses.

#### 3.2.8. Biosensors

Biosensors are analytical devices that use a biological component (such as enzymes, antibodies, etc.) to detect target molecules specifically, and the resulting signal is converted into a quantitative or qualitative measure of the concentration of those molecules. The biorecognition element plays a critical role in detecting viruses in biosensors, with antibodies or nucleic acids commonly used. Detection approaches include optical and electrochemical signals, as well as paper-based and nanotechnology-based devices [154]. Biosensors must adhere to the REASSURED criteria (Real-time connectivity, Ease of specimen collection, Affordable, Sensitive, Specific, User-friendly, Rapid and robust, Equipment-free or simple, Environmentally friendly, Deliverable to end-user) [155]. Paper-based microfluidics biosensors, while meeting some of these criteria, are primarily designed for immunoassays rather than nucleic acid detection. Altering physical support to control pore size and charge could enhance sensitivity, especially for wastewater monitoring [154]. A lot of advantages are worth noting: (a) biosensors can detect even small concentrations of viruses in wastewater samples due to the high specificity of the biological component used to recognize the target molecules; (b) virus detection can be achieved in a short time compared to other detection methods, allowing the rapid monitoring of wastewater quality; (c) biosensors can be integrated into automatic wastewater quality monitoring and control systems, allowing real-time data collection and interpretation; and (d) biosensors can be portable and miniaturized and can be used directly in the field for wastewater quality analysis without the need for sample processing in the laboratory. Despite all these advantages, the use of biosensors for the detection of viruses in wastewater still presents some challenges, especially regarding the pH, salt concentration, and/or temperature changes on antibody stability [156], such as the need to optimize and validate the devices for different types of viruses, as well as the development of standardized methods to evaluate the performance of biosensors under real field conditions. Another challenge is based on the changes in the epitopes used for detection due to changes in the viral genome [157]. Methods based on nucleic acid amplification are more difficult to use. DNA/RNA probes can be used as recognition elements to build hybridization-based biosensors, but the lysis of viruses and the extraction of viral DNA/RNA fragments is not direct, requiring a high temperature and chemical reagents [158], and hybridization/ nucleic acid amplification methods -require a complex thermal cycle. Alzate et al. (2022) developed a highly specific, sensitive, and portable electrochemical genosensor designed to detect HEV genotype 3 in wastewater in a sandwich-type format using specific DNA target probes that hybridize a sequence within the ORF2/ORF3 overlapping region of the HEV genome. The detection mechanism involves a biotinylated capture probe and a digoxigenin-labeled signal probe, with electrochemical detection facilitated by an anti-Dig antibody labeled with the horseradish peroxidase (HRP) enzyme [159]. Sukjee et al. (2022) introduced an electrochemical polymer composites biosensor for detecting SARS-CoV-2 particles in environmental samples. This biosensor features a sensitized layer made of molecularly imprinted polymer (MIP) composites, created using inactivated SARS-CoV-2. It demonstrated the capability to detect SARS-CoV-2 at concentrations as low as 0.1 fM in buffer solutions and reservoir water samples, maintaining linearity across a 3 log-scale range [160]. Kumar et al. (2021) developed a low-cost electrochemical DNA biosensor using printed circuit board (PCB) electrodes for detecting SARS-CoV-2 in wastewater [161]. The biosensor utilizes portable PCR instruments, such as miniPCR^®^, with the detection of PCR amplicons through methylene blue (MB) intercalation with DNA, increasing the voltammogram peak current at MB’s redox potential. The biosensor can detect SARS-CoV-2 nucleocapsid gene amplicons at concentrations as low as 10 pg/μL (approximately 1.7 fM) and identify nucleotides after 10 PCR cycles. These reusable electrodes are easy to clean, have a long shelf-life, and do not require surface modification [161].

The continued progress in biosensor technology may offer promising perspectives for improving wastewater quality monitoring and management. Biosensors should embody the four “M4” aspects: Modularity, Multiplexing, Multifunctionality, and Miniaturization. Modularity allows flexibility in detecting different viral targets, while multiplexing enables the detection of multiple targets simultaneously, including specific variants. Multifunctionality ensures biosensors are resistant to inhibitors or interference in wastewater, facilitating direct detection from raw samples. Finally, miniaturization reduces costs and enables portability, which is crucial for the deployment in sewage systems [154].

### 3.3. Virus Isolation in Cell Cultures

Cell culture virus isolation is a powerful technique used to quantify viable viruses from wastewater samples. The method is based on the principle that viruses can grow in sensitive cell lines in the laboratory. In contrast, the cytopathic effect, which appears as the deterioration of a monolayer of infected cells and is observed with light microscopy, is used for measuring infectious viruses [162]. One of the most significant advantages of using cell cultures in epidemiological studies is the ability to isolate and characterize infectious viral particles.

The most important limitation of this technique is that several viruses do not grow in cell culture, while others do not show visible cytopathic effects and, therefore, are not detected by conventional cell culture methods. A solution for these inconveniences was the development of an integrated cell culture/polymerase chain reaction technique. This method ensures obtaining results within 4–5 days of sewage-sample collection compared to the conventional method, which takes more than 18 days to provide the same results [163,164]. Furthermore, cells can be infected by various types of viruses that induce similar cytopathic effects, making it difficult to identify the specific viruses responsible for the observed one. An additional factor influencing the results is the competition between two viruses or viral variants for the same cell culture. A recent study regarding the Omicron and Delta variants of the SARS-CoV-2 infection in the cultures of human epithelial and lung cells showed that Omicron has a larger viral load in the human nasal cells [165]. This competition leads to selecting a single, dominant variant, eliminating the other variants [166]. On the other hand, many viruses can be found in wastewater at the same time, and, when they are cultivated, a cytopathic effect caused by another virus may appear, while other viruses would produce cytopathic effects later. These viruses will be lost during the in vitro passages. To overcome some of these limitations, cell culture methods have been combined with other diagnostic techniques to ensure the accuracy and reliability of the results obtained.

The isolation of viruses in cell cultures is essential for the study of viral infectivity, and, although it involves certain technical and logistical challenges, this method remains a mainstay in virological research and in the diagnosis of infectious diseases.

## 4. Wastewater Treatment and Proper Management

Viruses present in wastewater can threaten public health. Consumption or contact with contaminated water can lead to gastrointestinal, respiratory, or other ailments. Untreated or insufficiently treated sewage can contribute to the spread of disease in communities.

Wastewater is treated in sewage treatment plants to remove pollutants, including pathogenic micro-organisms. Chemical and biological treatment can help remove or inactivate viruses from wastewater so that they can be released into the environment in a safer and less polluting state, thereby reducing the risk of spreading disease. Several methods and technologies are used in wastewater treatment, and the procedure selection depends on the specific characteristics of the effluent, the desired quality level of the treated water, and the available resources.

Wastewater management, a critical process for public health, implies purification in wastewater treatment plants [167] and generally consists of several stages: preliminary treatment or pre-treatment (physical and mechanical); primary treatment (physicochemical and chemical); secondary treatment or purification (chemical and biological); tertiary or final treatment (physical and chemical); and treatment of the sludge formed (supervised tipping, recycling, or incineration) [168]. These stages are combined depending on inflow wastewater quality and the function of the treatment degree needed to meet acceptability standard [169]. As such, depending on the intended use of the effluent, for irrigation, recreational purposes, or discharge in rivers, tertiary treatment can be applied in order to eliminate pathogen micro-organisms [167].

Wastewater treatment ponds represent the most common wastewater primary treatment system [170]. The process of virus reduction by 1 log10 by this method needs between 14.5 to 20.9 days [171]. Virus removal in wastewater treatment ponds is determined by the combination of different factors like sedimentation, and sun-light mediated inactivation, as well as virus interaction with different particles or with other micro-organisms [170,171].

The sedimentation process is considered the major particle removal mechanism in wastewater treatment ponds. During this process, viral particles can be removed by adsorption to suspended solids and precipitation together with these [56,171,172]. Nevertheless, sedimentation does not completely remove viruses from wastewater [56].

Another commonly used wastewater treatment is sand filtration, which assures particle removal and, thus, virus particles adsorbed on their surface. However, they are not very efficient in virus removal [170]. Their removal capacity can be improved by adding different additives. Samineni et al. (2019) developed sand filters functionalized with the water extract of Moringa oleifera seeds that achieved an efficiency of 7 log10 MS2 bacteriophage, a surrogate for norovirus and rotavirus removal, as well as a regenerating procedure for it using saline water [173]. Sand-anthracite filters were reported to partially remove noroviruses during the tertiary treatment of wastewater [174].

Membrane filtration is a treatment process that assures wastewater decontamination using different types of filters. Depending on the pore size of the filter, membrane filtration is classified as microfiltration, ultrafiltration, nanofiltration, and reverse osmosis [170,172,175]. Reverse osmosis is commonly used to complete the removal of contaminants but can also reduce viruses in wastewater. In order to remove foulants that can interfere in the reverse osmosis process, including virus removal, it is usually combined with ultrafiltration as a pre-treatment system [170,176]. Moreover, reverse osmosis combined with the membrane bioreactor was reported to be efficient in norovirus removal [174]. Ultrafiltration is a low-cost alternative to reverse osmosis [175] and is regarded as the most effective way of virus removal [167]. Lee et al. (2017) evaluated the efficacy of the coagulation–sedimentation–ultrafiltration combination as a tertiary treatment system of wastewater on virus removal and determined that it is negatively correlated with the concentration of dissolved organic matter and can be improved by pH optimization to increase coagulation efficiency [177]. Moreover, in order to improve filtration efficiency, this process can be realized by combining low-pressure membrane filtration followed by high-pressure membrane filtration [178]. However, filtration and sedimentation are two processes that allow virus separation without their inactivation, which leads to the necessity to further inactivate viral particles in the resulting sludge [56].

Photocatalytic membrane reactors (PMRs) represent a combination of membrane filtration with photocatalysis [56]. They are a promising solution to remove viruses and other pathogens from water [179]. Guo et al. (2015) elaborated a PMR based on TiO_2_ tubular ceramic microfilters combined with UV disinfection and evaluated their efficiency in laboratory conditions on virus removal from high-turbidity feed water, using viable P22 bacteriophage as a surrogate. The efficiency of the tested system was better than stand-alone microfiltration, stand-alone UV disinfection, and microfiltration–UV with a non-photocatalytic membrane. Moreover, they concluded that virus removal and inactivation could be improved by the membrane pore size, the design of the photocatalytic coating, and controlling the UV fluence applied to the permeate stream [180]. There are several PMRs tested for wastewater treatment based on photocatalytic TiO_2_ films and membranes [181].

Ultraviolet (UV) disinfection photochemically damages DNA/RNA of the micro-organism [182,183]. Usually, it needs a short contact time and does not form by-products [56]. Pulsed UV irradiation and low-pressure UV irradiation inactivate F-specific RNA (FRNA) bacteriophage, used as a surrogate for norovirus, at high UV doses [182]. UV efficiency can be affected by the presence of suspended solids in secondary treated wastewater; thus, it is usually combined with a filtration system [182].

Viruses can also be removed by disinfection with different chemicals like chlorine, chlorine dioxide, ozone, chloramine, detergents, and alcohols [184], that act by oxidizing the protein layer or the structure of DNA/RNA [185]. Nevertheless, the chemical treatment of wastewater leads to the formation of residuals that need to be removed by additional treatments. Moreover, some viruses are resistant to the disinfection process [56].

Chlorination is one of the most applied chemical treatments of wastewater, being also the most successful method to inactivate viral particles. It is frequently used as a disinfection strategy for hospital sewage [186,187]. Chlorination is carried out using chloramines, sodium hypochlorite, chlorine dioxide, calcium hypochlorite, and chloroisocyanurates that generate hypochlorite ion (ClO^−^), one of the most powerful oxidizing agents, and hypochlorous acid (HOCl), a microbicidal agent [170,188]. Its efficiency is influenced by contact time, dose, pH, temperature [169], and wastewater composition (the presence of organic materials, ammonia, nitrites, etc.) [169,185,188]. Moreover, this type of treatment needs additional steps of the dechlorination and inactivation of toxic by-products in effluent water [185,188].

The ozonation mechanism is based on the formation of hydroxyl radicals that present a high reactivity [169]. Ozone inactivates viruses by destroying viral proteins, being effective against both enveloped and non-enveloped viruses [188,189,190]. Ozonation proved to be efficient in various virus removal [191,192]. Sigmon et al. (2015) determined that no more than 1.0 mg min/L is needed for the efficient inactivation (by 4 log10) of different pathogens from wastewater [192]. Ozone was reported to be used for the disinfection of sewage effluents used for crop irrigation or discharged to surface water [170].

Hydrogen peroxide is considered a non-toxic treatment since its by-products are not considered a health threat. As a disinfectant agent, it generates hydroxyl radicals that damage different viral components. Nevertheless, its activity is moderate and high amounts are needed for disinfection. As such, in order to increase hydroxyl radical generation, hydrogen peroxide is frequently combined with UV or ozone treatments [185]. An alternative option is a performic acid, a combination of hydrogen peroxide and formic acid, that is also efficient in virus inactivation, being superior to chlorination by quick decomposition and no toxic by-products [193,194].

Recent developments in the nanotechnological field offer the possibility to use different nanomaterials (e.g., nano biosorbents, nanocatalysts, bioactive nanoparticles, nanostructured catalytic membranes, nano bioreactors, and nanoparticle-enhanced filtration) in wastewater treatment in order to eliminate waterborne pathogens, including viruses [175,195]. For instance, nanometals inactivate viruses by generating ROS that cleave their DNA or RNA, while carbon-based nanomaterials are characterized by a large surface area and a high surface-area-by-volume ratio that assure them a high adsorbent capacity and facilitate virus removal [195]. Different systems like nanocomposite filter systems [196,197], Cu_2_O-coated multi-walled carbon nanotubes [198], and membranes [199] were developed and proven to be efficient in virus removal.

Biological methods of wastewater treatment are applied as secondary treatments. They are based on the activity of micro-organisms in order to oxidize organic materials from wastewater and include conventional activated sludge, aerated lagoons, oxidation ponds, membrane bioreactors, etc. [200,201]. The most used method, the activated sludge process, is reported to partially remove adenoviruses, polyomaviruses, rotaviruses [202], and noroviruses [174].

Membrane bioreactors (MBRs) represent activated the sludge biological treatment combined with membrane filtration [185,203]. Under optimal conditions, they proved efficient in removing human adenoviruses [203,204], enteroviruses [203,205], and noroviruses [203,204,205] due to the virus attachment to solid particles, and the retainment by the membrane or by the cake layer, as well as virus inactivation by predation or enzymatic breakdown [204]. Nevertheless, MBR use is associated with frequent membrane cleaning, as well as virus-contaminated sludge formation that requires decontamination procedures [185,206,207,208].

Microalgae and macroalgae are used in wastewater treatment as part of photobioreactors, biofilm reactors, MBRs, and oxidation ponds [56,206,209,210]. Virus inactivation in algae-based wastewater treatment systems is based on such mechanisms as sunlight-mediated inactivation, pH shift, and predation, as well as on virus attachment to biomass and sedimentation [56,206]. According to Delanka-Pedige, Cheng et al. (2020), the algal wastewater treatment system efficiency in reducing Enterovirus and Norovirus GI was comparable to that in the conventional wastewater treatment system with chlorination [211]. Moreover, heterotrophic nanoflagellates are efficient in the removal of virus-like particles [212]. Nevertheless, on the other side, Scheid and Schwarzenberger (2012) demonstrated that free-living amoebae from water reservoirs are able to act as carriers or vectors of adenoviruses, contributing to their transmission [213].

During the COVID-19 outbreak, there were several studies focused on the role of different wastewater treatment stages in reducing viral RNA concentrations. The study of Abu Ali et al. (2021) revealed that each of the primary and secondary treatment steps in two wastewater TP in Israel reduced SASR-CoV-2 RNA concentration only with 1 log, the viral RNA being removed only by chlorination (tertiary treatment). Moreover, they observed that an insufficient chlorine dose could lead to SARS-CoV-2 RNA detection after tertiary treatment [214]. Serra-Compte et al. (2021) evaluated the SARS-CoV-2 presence in wastewater before and after treatments, as well as in the resulting sludge. They detected viral RNA in 50.5% of the influent samples analyzed. The evaluation along water treatment lines revealed a significant reduction in viral RNA occurrence. The virus was detected in 23.3% of samples after secondary treatment and was not detected after the tertiary treatment step (MBR and chlorination). On the other side, the SARS-CoV-2 RNA was detected in non-treated sludges after both primary and secondary treatments. Moreover, viral RNA was also detected in the treated effluent by thickening and anaerobic digestion sludges, being removed only after thermal hydrolysis [215].

Regardless of technological development, current strategies in wastewater treatment suffer from different limitations like the formation of high by-product pollution, the high operating cost, the need for chemical additives, and the production of waste stream [167]. Moreover, viruses may still remain in the wastewater effluent since their removal from wastewater is influenced by different factors like their structure [56,216] and adsorptive behavior [217], wastewater pH [177,216], and temperature and composition [216]. Therefore, there is a need for alternative disinfection methods to eliminate pathogens from wastewater effectively.

Implementing appropriate wastewater management practices is crucial for preventing the spread of viruses and other pathogens. These may include properly collecting and treating wastewater, water disinfection, and educating the public about personal hygiene. Continuously monitoring wastewater quality and treatment is also important to ensure contamination levels are effectively controlled and to minimize health risks.

It should be noted that the chemical treatment of wastewater must be carried out carefully to avoid introducing harmful chemicals into the environment and to obtain treated water that is safe and suitable for discharge or reuse. The careful planning of chemical dosages, water quality monitoring, and adherence to safety standards are essential in this process.

It is important to understand that viruses in wastewater do not automatically mean that the water is dangerous. Proper wastewater treatment can significantly reduce contamination levels with viruses and other micro-organisms. In addition, proper personal hygiene, including hand washing, can help prevent the spread of infections associated with these viruses.

## 5. The Importance of Wastewater Surveillance for Public Health

Improper wastewater management and sanitation practices can be linked with new epidemic diseases, including zoonotic infections. Wastewater-based epidemiology, or sewage surveillance, has proven highly valuable for the early detection and surveillance of new epidemic diseases. For instance, it has been crucial in tracking poliovirus circulation and detecting SARS-CoV-2, the virus responsible for COVID-19, even before clinical cases are confirmed. This technique has also been applied to monitor norovirus outbreaks, antibiotic-resistant bacteria prevalence, and emerging pathogens. By providing early indicators of disease presence and trends in a community, wastewater surveillance empowers public health officials to take prompt actions to prevent the spread of diseases, safeguard public health, and allocate resources effectively.

Establishing robust disease surveillance systems requires the collaboration between health agencies, water authorities, and environmental agencies. The rapid reporting and sharing of information are crucial for preventing the spread of diseases. Addressing wastewater-related health challenges requires the collaboration between public health officials, environmental experts, engineers, and policymakers. Interdisciplinary efforts are needed to develop effective solutions.

Both the United States (US) and the European Union (EU) have regulations and legislations that play a critical role in wastewater management, specifically focusing on aspects related to environmental protection and public health. However, it is important to note that wastewater management regulations are primarily governed by environmental agencies, such as the US Environmental Protection Agency (EPA) in the United States and the European Environment Agency (EEA) in the European Union. Such key regulations and legislations include the Clean Water Act (CWA) [218] and Safe Drinking Water Act (SDWA) [219] in the USA, and The Urban Wastewater Treatment Directive (91/271/EEC) [220] and The Water Framework Directive (2000/60/EC) [221] in the EU. These regulations and legislation set standards for wastewater treatment, quality, and discharge, aiming to protect the environment, public health, and the well-being of communities.

## 6. Conclusions

Wastewater monitoring provides essential information about water quality and the degree of contamination. Monitoring these waters helps identify and manage risks to public health, prevent the spread of disease, and protect the environment. It is necessary to standardize the appropriate and most accurate methods for the isolation and identification of viruses in the effluent. Wastewater surveillance plays a crucial role in monitoring sewage systems and treatment plants’ proper functioning, as outlined by the CDC’s wastewater surveillance initiative. It involves assessing sewage systems’ performance in collecting and transporting wastewater and verifying the efficiency of treatment processes in treatment plants. Monitoring wastewater helps protect biodiversity, prevent water pollution, and positively impact the environment. It also identifies opportunities for water reuse and recycling, promoting the sustainable use of natural resources and reducing freshwater consumption. Ensuring treated water safety for discharge into the environment or reuse for purposes like agricultural irrigation or drinking water supply is essential. The viral surveillance of wastewater contributes to the early detection of infection outbreaks and the implementation of control measures to limit waterborne disease spread, particularly crucial in dense communities with an increased risk of transmission.

## Figures and Tables

**Figure 1 microorganisms-12-01430-f001:**
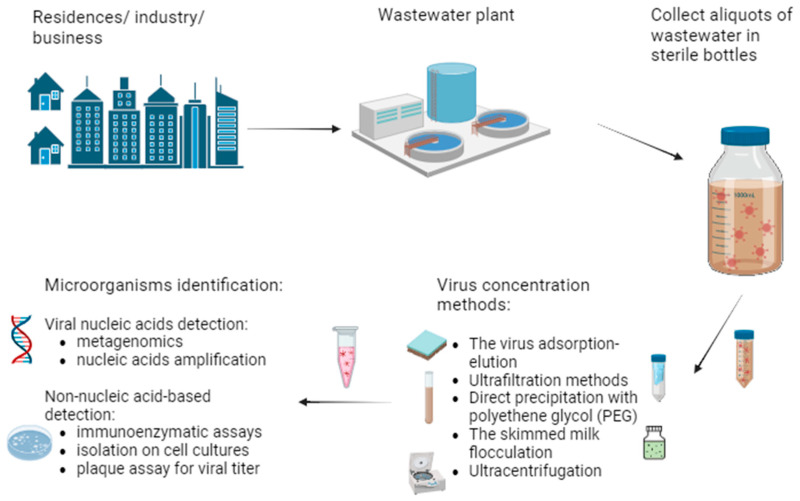
Workflows for virus detection in wastewater for environmental surveillance. Image was created with BioRender.com (https://app.biorender.com/, accessed in 1 June 2024).

**Table 1 microorganisms-12-01430-t001:** Main viruses that are frequently detected in wastewater and their prevalence.

Viruses	Associated Human Disease	Primary Transmission Route	Country	Prevalence (%)	Reference
Human adenoviruses (HAdV) dsDNA	pneumonia, respiratory tract infections, conjunctivitis, gastroenteritis	respiratory, fecal–oral	Taiwan	27.3	[14]
			Morocco	45.5	[15]
			Tunisia	64	[16]
			Poland	92.1	[17]
			Norway	92	[18]
Rotavirus (RV) dsRNA	gastroenteritis	fecal–oral	Uruguay	10	[25]
			Iran	25	[26]
			Thailand	50	[27]
			Spain	90	[28]
			Poland	60	[61]
Noroviruses (NoVs) ss RNA (+)	acute gastroenteritis	fecal–oral	Brazil Genotype I (GI) Genotype II (GII) Argentina GI GII Poland GI GII	38.5 96.1 1 14 73.3 80	[62] [63] [61]
Human enteroviruses (EVs) ss RNA (+)	gastroenteritis, conjunctivitis, herpangina, myocarditis, hepatitis, poliomyelitis, meningitis.	fecal–oral	France	100	[64]
			Italy	88.1	[65]
			Southwest United States	41.67	[66]
			Tunisia	35	[67]
Human hepatitis A virus (HAV) ss RNA (+)	hepatitis	fecal–oral	France	59.25	[64]
			Egypt	15	[68]
			Argentina	39	[69]
			Tunisia	66.9	[70]
Human hepatitis E virus (HEV) ss RNA (+)	hepatitis	fecal–oral	Argentina	22.5	[69]
			France	37	[64]
			Spain	30	[71]
			Switzerland	32	

**Table 2 microorganisms-12-01430-t002:** The major advantages and disadvantages to consider when choosing the appropriate method for the concentration of viruses in wastewater.

	Advantages	Disadvantages
Virus adsorption–elution (VIRADEL)	- Allows processing of large sample volumes (up to 100–1000 L); - Can be applied to various viruses, including those with different electrical charges; - Compatibility with various wastewater matrices; - Flexibility in the choice of adsorption materials (various materials for adsorption can be used, such as glass filters, nitrocellulose membranes, or quartz fibers); - Using electropositive filters increases the effectiveness of wastewater virus concentration by eliminating the need to adjust sample pH for virus adsorption; - Provides reproducible and consistent results under various experimental conditions.	- Increases the inhibitor concentration (due to the large volume); - Needs prolonged processing time (due to a multi-step process comprising pre-filtration, membrane adsorption, and elution); - Requires specific filters and elution devices, which can increase costs and the need for technical expertise; - The virus recovery efficiency can vary depending on the type of virus, the physical–chemical characteristics of the wastewater, and the type of electronegative/electropositive filter used; - Sensitivity to environmental conditions, pH, salinity, and the presence of organic or inorganic substances in the wastewater affects the technique’s performance; - Small variations in the procedure or environmental conditions can affect the reproducibility of the results; - May generate additional waste, such as used filters and elution solutions, which require proper management.
Ultrafiltration	- Allows size exclusion by using membrane filters with smaller pore sizes than viral particles; - High selectivity and efficiency, allowing the separation of viruses from other particles and impurities according to size and molecular weight, effectively concentrating the viruses; - Can process large volumes of water, being suitable for large-scale applications (up to several hundred liters, depending on the capacity and size of the equipment); - Protects the integrity of viruses, being a gentler method compared to ultracentrifugation; - Speed, reducing the total time required for virus concentration; - Reduced operating costs, consumables and equipment being more accessible; - There is the possibility of processing several samples simultaneously; - For samples with high turbidity, tangential flow ultrafiltration can be used, which allows water to flow parallel to the membrane surface.	- There is a possibility of clogging the filter with small pores when working with cloudy samples requiring frequent cleaning or replacement, or additional pretreatment steps are required to remove large particles and other contaminants that could block the membrane; - Specialized ultrafiltration systems and pumps may be required, which can complicate implementation; - Increasing the number of membranes due to clogging can increase costs; - Allows processing of small sample volumes (10–100 mL); - Virus recovery efficiency may vary depending on the type of membrane used and specific sample conditions; - There are limitations regarding pore size (selecting a membrane with the appropriate pore size is crucial; pores that are too large can allow viruses to pass through, while pores that are too small can reduce throughput and process efficiency); - Membranes and equipment require regular maintenance to maintain optimal performance and prevent clogging.
Direct precipitation with polyethylene glycol (PEG)	- Allows precipitation by sequestering water molecules from the solvation layer around viral capsids, enhancing virus–virus interactions; - High efficiency in recovering a wide range of viruses with good efficiency; - Compatibility with various wastewater matrices; - Relatively fast sample processing; - Procedural simplicity without requiring complex equipment; - Provides consistent and reproducible results; - PEG is chemically stable and can be stored for long periods of time without losing its effectiveness; - Is more suitable for small to moderate volumes of wastewater (up to 10 L).	- May cause non-selective precipitation of some inhibitors (chemical interferences), affecting recovery efficiency; - Requires careful handling to avoid sample loss; - Using PEG in large quantities can be expensive, especially for large volumes of wastewater, PEG being more expensive than other reagents; - It is more suitable for smaller volumes of wastewater (typically up to 50 mL–10 L), which may limit applicability to large-scale studies. The process requires careful handling and control of chemical parameters, making it difficult to apply to very large volumes; - Relatively long processing time involving several steps of precipitation and centrifugation; - Precipitated samples may require additional purification steps to remove PEG and other impurities before final analysis; - The precipitation efficiency can be influenced by the pH and temperature of the samples, requiring adjustments and optimizations of the experimental conditions.
Skimmed milk flocculation technique	- Requires minimal equipment and a reduced number of processing steps, facilitating the simultaneous concentration of numerous samples; - Easy to use and low-cost, requiring cheap and easily available materials; - Allows efficient recovery of viral particles, including those with different electric charges; - Is eco-friendly and does not require the use of dangerous chemicals; - Is scalable and can be applied to small and large wastewater volumes (up to 10–20 L).	- It has limited specificity so that not all types of viruses can be efficiently concentrated; - The presence of organic or inorganic substances can affect the efficiency; - The recovery rate of viruses can vary depending on the conditions of the wastewater; - Additional adjustments of pH or other parameters may be needed; - Residues may appear that complicate further analysis.
Ultracentrifugation	- Can be used for various viruses, regardless of size and molecular weight; - Simultaneously performs purification of other particles and impurities based on density and size; - Allows efficient and rapid recovery of viruses from wastewater samples; - Can concentrate viruses in a small volume of liquid, facilitating subsequent analysis; - Provides consistent and reproducible results under standard conditions; - The ultracentrifugation process can be rapid, reducing the total time required to concentrate viruses.	- High costs due to special ultracentrifugation equipment (high-speed ultracentrifuges, special rotors, and specific consumables) are expensive and require technical expertise and regular maintenance; - Efficiency can be affected by the initial conditions of the samples (chemical composition and presence of contaminants); - Limited scalability due to the volume of the centrifuge tubes and the high costs associated with equipment and consumables (up to 0.5–1 L); - The risk of degradation of some viruses that may be sensitive to high centrifugation forces, risking their degradation or inactivation.

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
