# Peer review of "Viruses in Wastewater—A Concern for Public Health and the Environment"

_microorganisms, 2024, doi:10.3390/microorganisms12071430_

Round 1
Reviewer 1 Report
Comments and Suggestions for Authors
The review "Viruses in Wastewater - a Concern for Public Health and the Environment" shows an updated revision of the different aspects involved for virus detection in wastewater, which is a growing trend in viral surveillance worldwide. However, this is not a novel review since there are already several reviews about this subject. I suggest to enrich this manuscript with more or different information in order to provide data that can be useful for the scientific community.
1.I suggest authors to include a table or graph that summarized the main viruses or viral families that can be more frequently detected in wastewater with their prevalence and concentration ranges.
2.Regarding the different concentrating methods for wastewater processing, it would be important to include a table that compares the advantages and disadvantages of these methods and which are the ones more commonly used nowadays.
3.I strongly suggest authors to incorporate a section in the manuscript describing the quantitative microbial risk assessment (QMRA) studies for the main pathogens detected in wastewater and its exposure risk reported according to this index.
4.Even though ultracentrifugation has several disadvantages I believe it is important to include a paragraph for this method in section 3.
5.Currently there are several NGS amplicon detection panels for respiratory and enteric viruses, this could be further described in the text.
6.The electron microscopy technique is not very useful for viral identification in wastewater since it is expected that more than one pathogen or phages can be present that can contaminate the sample, this should be discussed in the text.
7.I suggest to include an image summarizing the possible workflows options for virus detection in wastewater.
Author Response
The review "Viruses in Wastewater - a Concern for Public Health and the Environment" shows an updated revision of the different aspects involved for virus detection in wastewater, which is a growing trend in viral surveillance worldwide. However, this is not a novel review since there are already several reviews about this subject. I suggest to enrich this manuscript with more or different information in order to provide data that can be useful for the scientific community.
- Q: I suggest authors to include a table or graph that summarized the main viruses or viral families that can be more frequently detected in wastewater with their prevalence and concentration ranges.
A: Thank you for your suggestion. We included a table summarizing the main viruses that are frequently detected in wastewater and their prevalence.
- Q: Regarding the different concentrating methods for wastewater processing, it would be important to include a table that compares the advantages and disadvantages of these methods and which are the ones more commonly used nowadays.
A: Thank you for your suggestion. We included a table comparing the advantages and disadvantages of virus concentration methods.
- Q: I strongly suggest authors to incorporate a section in the manuscript describing the quantitative microbial risk assessment (QMRA) studies for the main pathogens detected in wastewater and its exposure risk reported according to this index.
A: Thank you for the suggestion and the opportunity to explain why using the QMRA index is useful in consumer risk assessment and treatment plant management.
- Q: Even though ultracentrifugation has several disadvantages I believe it is important to include a paragraph for this method in section 3.
A: Thank you for your suggestion and the opportunity to clarify this issue. Please find a paragraph regarding ultracentrifugation in section 3. “…..However, a study comparing two viral concentration methods, ultracentrifugation and skimmed-milk flocculation, for the detection of SARS-CoV-2 in wastewater samples demonstrated that the ultracentrifugation method displayed higher analytical sensitivity for the detection of enveloped viruses 36877444. Another study demonstrated that ultracentrifugation can be considered a suitable method to concentrate viruses directly from wastewater with a recovery percentage between 66 and 72% 22113738. However, in the case of this method, the efficiency of the results depends on the specific density, morphology and membrane attachment characteristics of each virus 20804786.”
- Q: Currently there are several NGS amplicon detection panels for respiratory and enteric viruses, this could be further described in the text.
A: Thank you for your suggestion and the opportunity to clarify these issues. We included a paragraph on this subject. “There are several NGS amplicon detection panels for respiratory and enteric viruses. For example,the Viral Surveillance Panel (Illumina) allows the characterization of more than 200 viruses and subtypes that are of high risk to public health, including SARS-CoV-2, Influenza, Mpox Virus, and Poliovirus. These panels are compatible with a range of samples, including wastewater and can detect both RNA and DNA viruses.”
- Q: The electron microscopy technique is not very useful for viral identification in wastewater since it is expected that more than one pathogen or phages can be present that can contaminate the sample, this should be discussed in the text.
A: Thank you for your suggestion. There are studies asserting that EM can be useful in the case of samples with mixed infections since the virus identification is made based on the morphology that is stable even after genetic mutations. “However, the main limitation of the method is the relatively high detection limit (107 particles/mL), requiring viral material to be concentrated. The main advantage of EM is the lack of specificity to any group of viruses, as opposed to immunological and molecular tests. With the present advances in genetic engineering and the use of a wide range of pseudovirus-based assays in vaccine development, the list of possible research targets for EM and the scope of problems solved have expanded significantly. In many cases, especially when it comes to mixed infections, electron microscopy turns out to be one of the most reliable virus detection tools. 33659809”
“This method allows the quantification, identification and classification of viruses according to morphology. TEM may be applied to identify viruses in emerging infectious diseases since viruses' morphologies are known to be stable even after the mutation of their nucleic acids 32758747”.
- Q: I suggest to include an image summarizing the possible workflows options for virus detection in wastewater.
A: Thank you for the suggestion. We have modified Figure 1 to summarize the workflows for virus detection in wastewater for environmental surveillance.
Reviewer 2 Report
Comments and Suggestions for Authors
The authors provided a review article on viruses in wastewater. The reviewer suggests to revise the manuscript as follows.
1) In Figure 1, "cell cultures" is advised to be replaced with "plaque assay with cell cultures", because viruses are not cells.
2) The subheading "3.1 Methods for Wastewater Viruses' Isolation" needs to be considered again, because the word "isolation" is often used for the microorganisms which can form colonies. The word “concentration” may be better reflecting the content of this section.
3) The subheading “3.2 Methods Spotlighting Viruses in Wastewater” needs to be considered again. “Spotlighting” is too broad word. In addition, this section is too long. The authors can separate this section into several sections. One of the candidates for the subheading is “PCR-based detection of viruses”.
4) The authors do not use consistent style of references. This journal employ the reference style of [**] and does not use the style of “** et al.”, although the authors at least use 16 times such as “Qiu et al.,”, “Sidhu et al.”, “Wong et al.”, “Wang et al. ”, “Palacios et al. [116]”, “Alcazate et al (2022)”, ” Sukjee et al. (2022)”, “Kumar et al. (2021)”, “Samineni et al.”, “Lee et al”, “Gou et al.”, “Sigmon et al.”, “Delanka-Pedige et al.,”, “Ali et al.”, and “Serra-Compte et al. [198]”. Please follow the uniform way of citation.
5) The subheading “5. The Significance of Proper Wastewater Management for Public Health” can be replaced with “5. The importance of wastewater surveillance for public health” because the original subheading can be understood as several meanings.
6) The reviewer suggests the inclusion of additional one or two tables, because this review article does not include any tables. Without tables, the readers cannot understand the overview of state and the art of the topic. For example, the table can list out target viruses, sampling location, concentration method, detection methods, and citation number.
Comments on the Quality of English Language
The reviewer did not notice fatal problems in the language.
Author Response
The authors provided a review article on viruses in wastewater. The reviewer suggests to revise the manuscript as follows.
1. Q: In Figure 1, "cell cultures" is advised to be replaced with "plaque assay with cell cultures", because viruses are not cells.
A: Thank you for your observation and the opportunity to clarify this issue. We have modified Figure 1 to specify that cell cultures are used for viral isolation and included the plaque assay that is used for viral titration.
2. Q: The subheading "3.1 Methods for Wastewater Viruses' Isolation" needs to be considered again, because the word "isolation" is often used for the microorganisms which can form colonies. The word “concentration” may be better reflecting the content of this section. and
3) The subheading “3.2 Methods Spotlighting Viruses in Wastewater” needs to be considered again. “Spotlighting” is too broad a word. In addition, this section is too long. The authors can separate this section into several sections. One of the candidates for the subheading is “PCR-based detection of viruses”.
A 2 and 3: Thank you for your suggestions and the opportunity to clarify this issue. We organised the chapter 3. Methods and Laboratory Techniques for Spotlighting Viruses in Wastewater in 3 sub-chapters as follows:
3.1. Methods for Wastewater Viruses’ Concentration.
3.2. Methods for Wastewater Viruses’ Identification
3.3. Virus isolation in cell cultures
4. Q: The authors do not use consistent style of references. This journal employ the reference style of [**] and does not use the style of “** et al.”, although the authors at least use 16 times such as “Qiu et al.,”, “Sidhu et al.”, “Wong et al.”, “Wang et al. ”, “Palacios et al. [116]”, “Alcazate et al (2022)”, ” Sukjee et al. (2022)”, “Kumar et al. (2021)”, “Samineni et al.”, “Lee et al”, “Gou et al.”, “Sigmon et al.”, “Delanka-Pedige et al.,”, “Ali et al.”, and “Serra-Compte et al. [198]”. Please follow the uniform way of citation.
A: Thank you for your observations. We have corrected the reference format accordingly. The number of citation [**] is indicated at the end of the corresponding phrase. The in-text citation of authors was uniform as Author et al. (year) in whole manuscript.
5. Q: The subheading “5. The Significance of Proper Wastewater Management for Public Health” can be replaced with “5. The importance of wastewater surveillance for public health” because the original subheading can be understood as several meanings.
A: Thank you for the suggestion and the opportunity to increase the clarity of the manuscript. We modified the title as you recommended.
6. Q: The reviewer suggests the inclusion of additional one or two tables, because this review article does not include any tables. Without tables, the readers cannot understand the overview of state and the art of the topic. For example, the table can list out target viruses, sampling location, concentration method, detection methods, and citation number.
A: Thank you for the suggestion. As recommended, we included in the manuscript a table that provides an overview of the state of the art of the topic, however, trying to be non-redundant with the text.
Round 2
Reviewer 1 Report
Comments and Suggestions for Authors
All the comments were addressed by the authors